# Farnesiferol C Exerts Antiproliferative Effects on Hepatocellular Carcinoma HepG2 Cells by Instigating ROS-Dependent Apoptotic Pathway

**DOI:** 10.3390/ph15091070

**Published:** 2022-08-28

**Authors:** Ahmed Alafnan, Abdulwahab Alamri, Jowaher Alanazi, Talib Hussain

**Affiliations:** Department of Pharmacology and Toxicology, College of Pharmacy, University of Ha’il, Ha’il 55476, Saudi Arabia

**Keywords:** farnesiferol C, HepG2 cells, ROS, apoptosis, intrinsic apoptotic pathway, caspase-3

## Abstract

Farnesiferol C (Far-C) is a coumarin commonly extracted from *Ferula asafetida* and is popularly used as a traditional source of natural remedy. Liver cancer or hepatocellular carcinoma (HCC) has emerged as a major cause behind cancer burden, and limited therapeutic interventions have further aggravated the clinical management of HCC. In the present study, the authors tested the hypothesis that Far-C-instigated oxidative stress resulted in anti-proliferation and apoptosis instigation within human liver cancer HepG2 cells. The observations reported herewith indicated that Far-C exerted considerable cytotoxic effects on HepG2 cells by reducing the cell viability (*p* < 0.001) in a dose-dependent manner. Far-C exposure also resulted in enhanced ROS production (*p* < 0.01) which subsequently led to loss of mitochondrial membrane potential. Far-C-instigated oxidative stress also led to enhanced nuclear fragmentation and condensation as revealed through Hoechst-33342. These molecular changes post-Far-C exposure also incited apoptotic cell death which concomitantly led to significant activation of caspase-3 (*p* < 0.001). Furthermore, Far-C exhibited its competence in altering the expression of genes involved in apoptosis regulation (*Bax*, *Bad*, and *Bcl2*) along with genes exerting regulatory effects on cell cycle (*cyclinD1*) and its progression (*p21^Cip1^* and *CDK4*). The evidence thus clearly shows the preclinical efficacy of Far-C against HepG2 cells. However, further mechanistic investigations deciphering the alteration of different pathways post-Far-C exposure will be highly beneficial.

## 1. Introduction

According to latest reports, 905,677 patients of liver cancer were reported from both sexes and all age groups, which constituted 4.7% of the total 19,292,789 cancer patients globally. Moreover, 830,180 liver cancer-associated mortalities were also reported globally during 2020, which constituted 8.3% of total 9,958,133 cancer-related mortalities [1]. Hepatocellular carcinoma, abbreviated as HCC, is reported to be involved in approximately 90% of all the diagnosed liver-associated malignancies and is predicted to affect >1 million individuals by 2025 [2]. During last couple of decades, a steep incline in cases of HCC have been reported from most European countries, American continents, and the Indian sub-continent [3]. Of all the other risk factors involved in the onset and progression of HCC, infection with hepatitis B (HBV) and/or hepatitis C (HCV) falls under the high-risk category [4]. At present, the treatment of HCC is dependent upon the stage at which the cancer is diagnosed, and therapeutical regimes may include organ transplant and/or surgical resection [5]. Sorafenib is the only systemic chemotherapeutic available for HCC patients with advanced stages; however, recently alternative therapeutical modalities were further approved. Intriguingly, due to a lack of efficacy of these modalities during clinical trials, exploration of new therapeutics remains a major concern for clinical management of HCC [6,7].

Since ancient times, plants have been a key constituents of several folk medicines due to their valued pharmacological properties [8]. These pharmacological attributes are directly associated with the presence of different bioactive phyto-compounds and one such group among these biologically active compounds is sesquiterpene coumarins. Coumarins, or 2H-1-benzopyran-2-one, are constituted by a group of phenol-based phyto-compounds formed by the fusion of α-pyrone and benzene rings [9]. Coumarin and its derivatives have previously exhibited potent inhibitory effects on proliferation of different cancer cells in vitro [10,11]. Moreover, coumarin derivative also showed its anti-metastasis efficacy in animal models [12,13]. Far-C is a representative of sesquiterpene coumarin found naturally within the roots of members belonging to the Ferula (Apiaceae) genus including *F. assa-foetida* and *F. szowitsiana* among others as shown in Figure 1. Far-c holds a unique chemical structure when compared with other sesquiterpene coumarins and thus is related with several biological attributes [14]. It is used as a food spice in many Asian countries and for the treatment of asthma, bronchitis, ulcer, kidney stone, pain, and cancer in traditional herbal medicine. *F. assa-foetida* L. was reported to have antitumor, antimutagenic, and antiviral activities [15]. The underlying mechanism of its antitumor activity and the active chemicals remain unclear. Cancer chemotherapeutics often exert their therapeutical effects by inducing oxidative stress such as augmenting the levels of intracellular reactive oxygen species (ROS) and other free radicals [16]. Therefore, in the present study, the authors tested the hypothesis that Far-C may instigate ROS-mediated apoptotic pathways in HCC HepG2 cells. The reported work explores the cytotoxic and anti-proliferative attributes of Far-C against human liver hepatoma HepG2 cells in vitro.

## 2. Results

### 2.1. Far-C Exerted Cytotoxic Effects on Liver Cancer Cells

MTT assay-based investigation was undertaken to assess the cytotoxic and anti-proliferative effects of Far-C against HepG2 cells after 24 h. Far-C exposure reduced the cellular viability of HepG2 cells which in turn was calculated to be 83.13 ± 2.65 (15 µM), 63.75 ± 4.58 (30 µM), and 34.40 ± 2.50 (60 µM) in comparison with untreated HepG2 control cells (Figure 2A). The inferences from MTT assay showed that Far-C exhibited significant cytotoxic efficacy against human liver cancer HepG2 cells.

Morphological evaluation of HepG2 cells post exposure with Far-C explicitly showed the presence of altered HepG2 morphology. The photomicrographs (Figure 2B) showed presence of shrunken, circular morphology following a dose-dependent trend. HepG2 cells also showed circular morphology (red arrow) and disintegrated organelles (orange arrows and blue arrows). These observations indicated that Far-C exerted cytotoxic effects resulted in morphological alterations in HepG2 cells in comparison with untreated control cells.

### 2.2. Far-C Induced Abruptions in Nuclear Morphology

The activation of apoptotic pathways is marked by onset of evident changes within the nuclear morphology and includes its condensation and fragmentation. Hoechst-33342 stain was used to qualitatively evaluate whether Far-C mediated toxic effects on HepG2 cells resulted due to the activation of apoptotic pathways. Far-C exposure instigated changes in the nuclear morphology proportionally with its concentration. The photomicrographs obtained after Hoechst-33342 stain showed increased nuclear condensation within HepG2 cells in comparison with untreated HepG2 control cells (Figure 3).

### 2.3. Far-C Exposure Induced Apoptosis within Liver Cancer HepG2 Cells

The apoptotic cell death induced by Far-C was also investigated by using the acridine orange/ethidium bromide (AO/EtBr) dual staining. AO/EtBr is used in assessing the nuclear morphology of apoptotic cells. AO is a vital dye that will stain both live and dead cells, whereas EtBr will stain only those cells that have lost their membrane integrity. As seen in Figure 4A, cells stained green represent viable cells (ViCs), whereas reddish or orange staining represents late apoptotic cells (ApCs). In the untreated control, uniformly green live cells with normal and large nucleus were observed, whereas in Far-C-treated cells, orange and red staining was observed. Far-C at maximum concentration of 60 µM instigated considerable blebbing and nuclear condensation both of which serve to be specific features associated with apoptosis. As shown in Figure 4B, dose-dependent decrease in the number of ViCs and increase in the number of ApCs was observed in comparison with untreated control HepG2 cells. Thus, these observations lead the investigators to conclude that Far-C exposure induced significant apoptosis in human liver cancer HepG2 cells.

### 2.4. Far-C Instigated Intracellular ROS and Dissipation of ΔΨm

Increased level of intracellular ROS is considered an important factor for activation of apoptotic pathways [17]. Thus, to gain a deeper insight into the mechanics of Far-C-mediated toxicity on HepG2 cells, levels of intracellular ROS were qualitatively assessed in Far-C treated/untreated HepG2 cells. The DCF-DA fluorescent photomicrographs clearly indicated that Far-C exposure led to escalated ROS production as seen by the increasing DCF-DA-mediated green fluorescence. As seen in Figure 5, ROS production was also found to be proportionally dependent on Far-C concentration.

The onset of mitochondria-dependent apoptosis is characteristically recognized by the loss in mitochondrial viability which in turn results due to the dissipation of ΔΨm. To reaffirm the efficacy of Far-C in inducing the dissipation of ΔΨm, mitochondria and its voltage-specific Rh-123 stain was used. The fluorescent photomicrographs, as shown in Figure 5, indicated the intensity of Rh-dependent fluorescence reduced proportionally with increase in the Far-C concentration. These observations clearly indicated that exposure of HepG2 cells to Far-C led to a considerable decline in mitochondrial viability in a dose-dependent manner.

### 2.5. Far-C Exposure Activated Caspase-3 and -9

Cysteine proteases such as caspases are important mediators of apoptotic cell death and are also involved in regulating the proteolytic cleavage of cellular proteins. Therefore, in the present study, we further examined the role of caspase during apoptotic cell death in HepG2 cells instigated by Far-C. Caspase-9 is involved in activation of intrinsic cell death pathway whereas caspase-3 is the main downstream executioner caspase. As shown in Figure 6A, a significant change in the activation levels of stated caspases was observed in HepG2 cells post-Far-C exposure after 24 h. The activity of caspase-3 was considerably increased to 59.69% ± 4.15% (15 µM), 97.46% ± 2.80% (30 µM), and 117.20% ± 3.22% (60 µM) in comparison with untreated control HepG2 cells. Moreover, caspase-9 activity was also found to be increased to 48.31% ± 4.80% (15 µM), 59.06% ± 4.42% (30 µM), and 79.25% ± 3.75% (60 µM) in comparison with untreated control HepG2 cells. Thus, these observations of increased caspase activities outlined the involvement of intrinsic apoptotic pathway in Far-C-mediated apoptosis in HepG2 cells.

### 2.6. Caspase Inhibitors Ameliorated Far-C-Instigated Apoptosis in HepG2 Cells

In order to reaffirm the involvement of caspases in Far-C-instigated cytotoxicity in HepG2 cells, MTT assay was carried out in human liver cancer HepG2 cells pretreated with Z-LEHD-FMK and ZDEVD-FMK (50 µM each for 2 h) which are specific inhibitors of caspase-9 and caspase-3, respectively. As per the observations, caspase inhibitors considerably ameliorated Far-C-instigated cytotoxicity in HepG2 cells (Figure 6B,C). Thus, it could be concluded by saying that the activation of caspases was indispensable in Far-C-mediated apoptotic cell death within human liver cancer HepG2 cells. However, caspase inhibitors failed to completely ameliorate Far-C-mediated cytotoxicity suggesting that pathways to other caspase activation may have also contributed during Far-C-mediated apoptotic cell death in HepG2 cells.

### 2.7. Far-C Treatment Modulated the Expression of Apoptotic and Cell Cycle Regulatory Genes

qRT-PCR was performed to assess the modulatory effects of Far-C on apoptotic and cell cycle progression-related gene expression. The expression levels of target genes mRNA were reported as fold change relative to control by 2ΔΔCT method. As shown in Figure 7, the mRNA expression of anti-apoptotic gene *Bcl2* was considerably reduced with a concomitant increase in the mRNA expression of pro-apoptotic Bax and Bad genes. The reduction in the mRNA expression of *Bcl2* gene was estimated to be 0.90 ± 0.03 (15 µM), 0.72 ± 0.05 (30 µM), and 0.51 ± 0.04 (60 µM) fold in comparison with untreated control HepG2 cells. The mRNA expression levels of *Bax* and *Bad* were also estimated to be increased by 1.46 ± 0.05 (15 µM), 1.95 ± 0.07 (30 µM), and 2.16 ± 0.08 (60 µM) and 1.45 ± 0.06 (15 µM), 1.71 ± 0.06 (30 µM), and 2.07 ± 0.02 (60 µM), respectively, compared with untreated control HepG2 cells.

The genes primarily involved in cell cycle progression namely *cyclinD1*, *CDK4*, and *p21^Cip1^* were also evaluated. *Cyclin D1* is an important protein mediating the regulation of G1 transition during cell cycle. As shown in Figure 7, post Far-C exposure, the expression of *cyclinD1* mRNA was reduced by 0.91 ± 0.02 (15 µM), 0.61 ± 0.04 (30 µM), and 0.39 ± 0.04 (60 µM) compared with untreated control HepG2 cells. Subsequently, the effect of Far-C treatment on expression levels of *p21^Cip1^* was also assessed and the results showed a dose-dependent increment. The *p21^Cip1^* mRNA was found to be at increased expression by 1.54 ± 0.04 (15 µM), 1.94 ± 0.06 (30 µM), and 2.28 ± 0.04 (30 µM) compared with the untreated control. Eventually, the effect of Far-C exposure on *CDK4* (cell cycle regulatory gene) was also assessed using qRT-PCR. As shown in Figure 6, a dose-dependent decrease in mRNA expression of *CDK4* was observed by 0.88 ± 0.04 (15 µM), 0.58 ± 0.03 (30 µM), and 0.35 ± 0.03 (60 µM) in comparison with the untreated control.

## 3. Discussion

During the past decade substantial success has been achieved in the prevention of HCC through increased HPV-B and HPV-C vaccination [18]. Presently, the conventional regime for clinical management of HCC is constituted by chemo- and/or radiotherapeutics, which in turn exert considerable adverse side effects including increased reports of drug resistance and systemic cytotoxicity. This scenario shows the need for developing novel therapeutic modalities which are less toxic and can act as adjunct/complementary medicine. Due to their intrinsic pharmacological attributes, coumarins have been considered as a group of bioactive compounds whose anticancer potential is not yet fully explored. Previously reported studies have elucidated that coumarins hold anti-proliferative potential in lymphocytic leukemia, colon, and hepatoblastoma in vitro [19]. Moreover, to date, there remains sparse information on anti-proliferative and apoptotic effects of Far-C against HCC HepG2 cells. Therefore, the investigators hypothesize that Far-C exposure could induce ROS-mediated apoptotic cell death in HCC HepG2 cells.

During the initial screening, it became evident that Far-C exposure resulted in significant cytotoxic effects on HepG2 cells. The cytotoxic effect resulted in reduced cellular viability of HepG2 cells following a dose-dependent trend. Moreover, the cytotoxic effects of Far-C also became evident during the morphological assessments. HepG2 cells exposed to varying Far-C concentration exhibited constricted cellular morphology along with increased number of detached cells. Intriguingly, Far-C failed to exert any such cytotoxic effects against normal murine alveolar macrophage cells.

Apoptotic pathways are crucial for maintain homeostatic environment and overall development of multicellular organisms. Importantly, homeostatic functioning of apoptotic pathways is also indispensable for naturally removing cancer cells and thereby impeding the proliferation of metastatic cancers. Owing to these roles, instigation of apoptosis has emerged as a frontline therapeutic target for clinical management of different carcinomas [20,21]. The fluorescent photomicrographs of Hoechst-33342-stained HepG2 cells pointed to the onset of apoptotic cell death due to the presence of abrupt nuclear morphology which included augmented levels of nuclear condensation. Several natural compounds have been explored recently as an alternative to current chemotherapeutics owing to their efficacy in instigating apoptosis in cancer cells [22]. Nevertheless, evading apoptosis remains to be a crucial strategy adopted by nearly every cancer cell which eventually results increased proliferation and metastasis of cancer cells [23]. During our investigation, Far-C showed its efficacy in instigating apoptosis by leading to the formation of apoptotic bodies in HepG2 cells. These apoptotic bodies were clearly seen in the AO/EtBr dual-stained fluorescent photomicrographs. These morphological alterations were also quantified using Image J software (NIH, Maryland, USA; v1.46r) and showed the efficacy of Far-C in instigating a dose-dependent onset of apoptosis in HepG2 cells.

Caspases are now regarded as a well-established member of cysteine protease family which regulates apoptosis [24]. Caspase-9 is considered to the main effector of mitochondrial-dependent/intrinsic apoptotic pathway whereas caspase-3 activation further pacifies the apoptosis by promoting the proteolytic cleavage of different cellular proteins [25]. The results obtained during assessment of caspase activities post Far-C exposure explicitly showed that activities of both the caspase were enhanced which may further be attributed to contributing to the apoptosis-inducing potential of Far-C against HepG2 cells. Furthermore, pre-treatment of HepG2 cells with specific caspase inhibitors ameliorated the cytotoxic potential of Far-C which again may be implicated to the reduced levels of apoptosis induced in the presence of specific caspase inhibitors. Therefore, it becomes rational to conclude that Far-C treatment induced the activation of caspase involved in mediating mitochondrial-dependent apoptotic pathway.

ROS serves to be the byproduct of oxidative stress which results from unbalanced generation and elimination of oxidant species. It is reported to be highly unstable and includes singlet oxygen, hydrogen peroxide (H_2_O_2_), hydroxyl radical, and superoxide anion radical. Furthermore, it has now been well-established that enhanced production of intracellular ROS exerts detrimental effects on proteins, lipids, and nuclear content within the cells and thus results in the induction of apoptotic cell death. ROS-mediated apoptotic cell death is mediated by the dissipation of ΔΨm which further aids the release of apoptotic factors in the cytoplasm [26]. The evidence presented in this report clearly established the efficacy of Far-C in instigating the ROS augmentation which was also coupled with dissipation of ΔΨm, both of which can plausibly be correlated with instigation of the mitochondrial apoptotic pathway.

In different forms of cancer, the advancement of cancer cells through different phases of cell cycle is tightly regulated by cyclin proteins and cyclin-dependent kinases (CDKs). Intriguingly, the inhibitors of cyclin-dependent kinases or CDKIs negatively regulate the progression of cancer cells through different phases of cell cycle. Interaction of *cyclinD1* with *CDK4* and *CDK6* during G1 phase modulates G1/S phase transition and thereby increases the cell proliferation [27]. The qRT-PCR-based evidence showed that Far-C mediated the reduced mRNA expression of both cyclinD1 and CDK4 in HepG2 cells, which has previously been reported to be responsible for arresting the progression of cancer cells to the G0/G1 phase of cell cycle. Subsequently, CDKIs including *p21^Cip1^* and *p27^Kip1^* are commonly associated with blocking the progression of cancer cells to the S phase of the cell cycle [28]. These CDKIs inhibit the kinase activity of cyclins/CDKs complexes by competing for binding with CDKs and preventing their interaction with cyclins. During the investigation, it was evident that expression of *p21^Cip1^* mRNA was considerably augmented which may also be correlated with plausible G0/G1 phase cell cycle arrest of HepG2 cells post Far-C exposure. Considerable scientific literature has now proven that *Bcl-2* family members are closely related and regulate the apoptotic pathways. Both the pro- and anti-apoptotic genes such as *Bax*, *Bcl-2*, and *Bad* are important for maintaining the homeostatic environment within a cell [29]. Far-C showed its efficacy in enhancing the expression of pro-apoptotic genes (*Bax*, *Bad*) with concomitant reduction in the mRNA expression of anti-apoptotic (*Bcl-2*) gene. Therefore, our reports provide a mechanistic insight into the ROS-mediated anti-cancer efficacy of Far-C against HCC HepG2 cells.

## 4. Materials and Methods

Farnesiferol-C was procured from Sigma Aldrich, USA. MTT dye (3-(4,5-dimethylthiazol-2-yl)-2,5-diphenyl tetrazolium bromide), Minimum Essential Media (MEM), iPurATM Total RNA Miniprep Purification Kit, RNase A, antibiotic–antimycotic solution, fetal bovine serum (FBS), acridine orange, and ethidium bromide were purchased from Himedia, India. Rhodamine (Rh)-123 and Hoechst 33342 were obtained from Sigma (St. Louis, MO, USA).

### 4.1. Cell Culture and Maintenance

Human liver cancer hepatoma G2 (HepG2) liver cells were commercially procured from National Centre for Cell Sciences (NCCS), Pune, India. The cells were incubated in a humidified environment constituting 5% of CO_2_ at 37 °C. The cells were allowed to proliferate in Minimum Essential Media (MEM) mixed with 1% (*v*/*v*) antibiotic-antimycotic solution and fetal bovine serum (10% *v*/*v*).

### 4.2. Cytotoxicity and Morphological Evaluations

The cytotoxic effects of Far-C against HepG2 cells was estimated using MTT assay, described previously [30]. Approximately, 5 × 10^3^ HepG2 cells were allowed to adhere in each well of a 96-well plate under standard culture condition. Post-adherence, these cells were exposed to varying concentrations of Far-C (15, 30, and 60 µM) for 24 h. Subsequently, the media in each well was decanted and the wells were supplemented with 10 µL MTT (5 mg/mL) and the plate was incubated at 37 °C for another 4 h. Thereafter, 100 µL DMSO was further supplemented in each well and the plate was left for 30 min at 37 °C in darkness. Finally, the absorbance of solubilized formazan crystals was recorded at 570 nm (Bio-Rad spectrophotometer, Hercules, CA, USA). Far-C instigated cytotoxicity on HepG2 cells was expressed in terms of cellular viability percentage and was calculated as
$$\mathrm{Cellular}\;\mathrm{viability}\;\%=\frac{Absorbance\;of\;Treated\;cells\;}{Absrobance\;of\;Untreated\;control\;cells}\times 100.$$

The morphology of HepG2 cells post-exposure with Far-C cells was visualized using microscope. The same procedure as mentioned above was repeated and HepG2 cells were exposed to 15, 30, and 60 µM Far-c and incubated for 24 h. Post incubation, the changes in the morphology of HepG2 cells were visualized under bright light of Floid imaging station (Thermo-Fischer Scientific, Waltham, MA, USA).

### 4.3. Assessment of Nuclear Morphology

A total of 5 × 10^3^ HepG2 cells were allowed to adhere overnight in each well of a 96-well plate and were subsequently exposed to varying Far-C concentrations (15, 30, and 60 µM). Post incubation, media in each well was replaced and the cells were treated using Hoechst-33342 as previously described [31]. Eventually, the fluorescent blue nuclei was visualized and captured using Floid imaging station (Thermo-Fischer Scientific, Waltham, MA, USA) at excitation:emission of 390/40:446/33 nm.

### 4.4. Evaluation of Intracellular ROS

ROS-mediated oxidative stress in Far-C-treated HepG2 cells was evaluated using DCFH-DA stain through a microscope as reported previously [32] with subtle alterations. A total of 1 × 10^5^ HepG2 cells/well were allowed to adhere overnight in a 96-well plate. Post adherence, the cells were exposed to varying concentrations of Far-C (15, 30, and 60 µM) for 6 h. Subsequently, the media in each well was decanted and replaced with 20 µM DCFH-DA and the plate was further incubated in darkness for 30 min at 37 °C. Finally, the cells were visualized for their DCF-DA-mediated green fluorescence using Floid imaging station (Thermo-Fischer Scientific, Waltham, MA, USA).

### 4.5. Evaluation of Mitochondrial Membrane Potential (ΔΨm)

The mitochondrial viability within HepG2 after exposure with Far-C was estimated using Rhodamine (Rh)-123 stain as reported previously [33]. A total of 5 × 10^3^ HepG2 cells were seeded in each well of a 96-well plate and allowed to adhere overnight under standard cell culture environment. Thereafter, HepG2 cells were subjected to varying Far-C concentrations (15, 30, and 60 µM) for 24 h. After incubation, the wells were decanted and re-supplemented with Rh-123 (5 mg/mL) for an additional 30 min at 37 °C in darkness. Finally, the green, fluorescent HepG2 cells were visualized, and the photomicrographs were captured using FITC channel of Floid imaging station (Thermo-Fischer Scientific, Waltham, MA, USA).

### 4.6. AO/EtBr Dual Staining

The initiation of apoptotic cell death within HepG2 cells exposed to varying concentrations of Far-C was assessed through acridine orange (AO) and ethidium bromide (EtBr) as previously described with subtle modifications [34]. Around 5 × 10^5^ HepG2 cells were allowed to adhere overnight in each well of a 96-well plate under ambient culture condition. HepG2 cells were then exposed to 15, 30, and 60 µM Far-C for 24 h. Subsequently, Far-C-exposed cells were detached and pelleted (1500 rpm; 2 min at 4 °C). The pellet was then resuspended in equi-volume (100 µL each) of AO and EtBr and briefly incubated at RT for 15 min. Eventually, the cells were visualized, and images were captured through red and green filters of Floid imaging station (Thermo-Scientific, Waltham, MA, USA).

### 4.7. Estimation of Caspase Activity

Activities of caspase-3 and -9 in Far-C-treated HepG2 cells were estimated using colorimetric kits as per the manufacturer’s instructions. Approximately, 3 × 10^6^ Far-C (15, 30, and 60 µM)-treated HepG2 cells were subjected to ice-cold 50 µL chilled lysis buffer briefly for 10 min on ice. The obtained suspension was subjected to centrifugation for 1 min at 10,000 rpm (4 °C). Post centrifugation, the supernatant was collected and kept on ice. Subsequently, equi-volume (50 µL in each well) of cell lysate and reaction buffer where the latter was constituted by 10 mM DTT. Thereafter, 4 mM DEVD-pNA substrate was supplemented in each well and the reaction was briefly incubated (10 min). The absorbance of each well was then recorded at 405 nm. Alteration in caspase-3 and -9 activities was expressed in terms of percentage (%) change by comparing it with that of untreated control HepG2 cells.

### 4.8. Effect of Caspase Inhibitor

The cytotoxic effects of Far-C against HepG2 were re-evaluated using caspase-3 and -9 inhibitors. HepG2 cells were pre-exposed to 50 µL of Z-DEVD-FMK and Z-LEHD-FMK for 2 h (caspase-3 and -9 inhibitor), respectively. After incubation, HepG2 cells were exposed to 15, 30, and 60 µM Far-C for another 24 h. Finally, the cellular viability of HepG2 cells was calculated through MTT assay as per the protocol mentioned in cytotoxicity evaluation section.

### 4.9. Quantitative Real-Time PCR

A total of 1 × 10^6^ HepG2 cells were allowed overnight adherence in each well of a 6-well plate under optimum culture condition. The cells were exposed to 15, 30, and 60 µM Far-C and allowed 24 h of incubation under optimum culture environment. The treated cells were subsequently used for isolating for total RNA using HiPurATM Total RNA Miniprep Purification Kit (Himedia, Maharashtra, India). Isolated RNA (2 µg) was used for synthesizing cDNA using the primers as mentioned in Table 1 through Verso cDNA synthesis kit (Thermo-Scientific, Waltham, MA, USA). Finally, qPCR was done using SYBR Green qPCR Kit (Thermo-Scientific, Waltham, MA, USA) as per the supplier’s manual. Normalization of the evaluated genes was made against glyceraldehyde 3-phosphate dehydrogenase (GAPDH). The data were calculated using comparative CT method and the fold change was estimated through 2ΔΔCT method.

## 5. Conclusions

In conclusion, our report showed that Far-C possesses a significant apoptosis-inducing potential against HCC HepG2 cells in vitro. The results obtained during the present investigation indicated that the anti-cancer potential of Far-C against HepG2 cells can be correlated with the instigation of the apoptotic pathway. The onset of apoptosis was further linked with enhanced ROS levels which could have played an important role in dissipation of ΔΨm, caspase activation, and modulation of genes involved in cell cycle progression and apoptosis. Thus, the report presented here describes the mechanism of Far-C-mediated anti-cancer effects against HepG2 cells.

## Figures and Tables

**Figure 1 pharmaceuticals-15-01070-f001:**
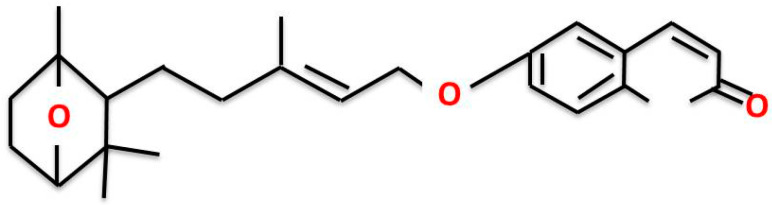
Structure of farnesiferol-C (Far-C).

**Figure 2 pharmaceuticals-15-01070-f002:**
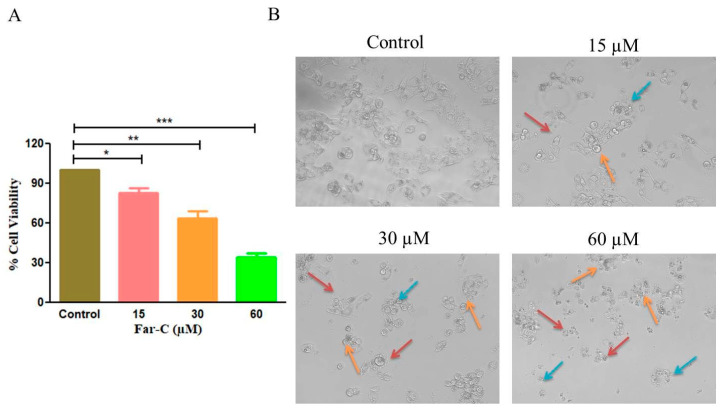
Cytotoxic effects of Far-C against human liver cancer HepG2 cells. (**A**) Cellular viability percentage of HepG2 cells and (**B**) changes in the morphological features such as circular morphology (red arrow), disintegrated organelles (orange arrows and blue arrows) of HepG2 cells post-24 h exposure with Far-C. Scale bar = 100 µm. * *p* < 0.05, ** *p* < 0.01, *** *p* < 0.001.

**Figure 3 pharmaceuticals-15-01070-f003:**
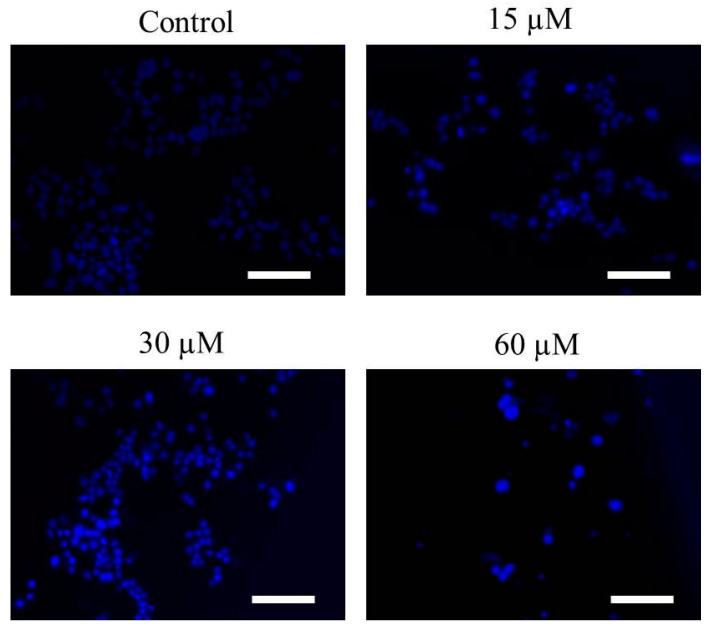
Changes in nuclear morphology observed through Hoechst-33342 staining in HepG2 cells post-24 h exposure. Scale bar = 100 µm.

**Figure 4 pharmaceuticals-15-01070-f004:**
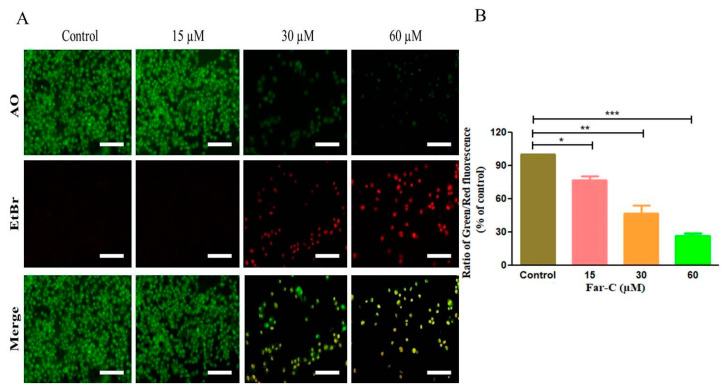
(**A**) Instigation of apoptotic cell death in HepG2 cells after treatment with Far-C (15, 30, and 60 µM) for 24 h and (**B**) quantification of apoptotic cell death through ImageJ software NIH, Maryland, USA, (v4). Scale bar = 100 µm. * *p* < 0.05, ** *p* < 0.01, *** *p* < 0.001.

**Figure 5 pharmaceuticals-15-01070-f005:**
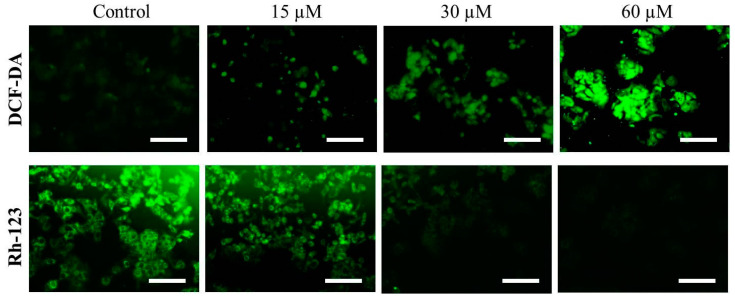
Efficacy of Far-C in augmenting the intracellular level of ROS and dissipation of ΔΨm in a dose-dependent trend. Scale bar = 100 µm.

**Figure 6 pharmaceuticals-15-01070-f006:**
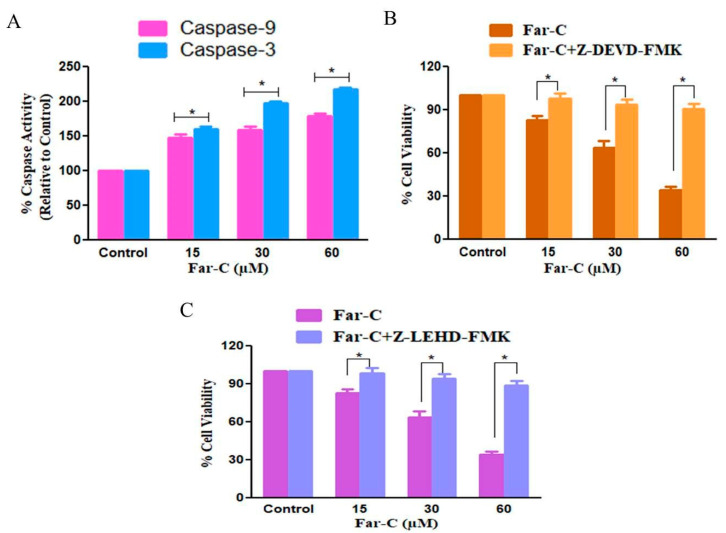
(**A**) Activation of different caspases post-exposure with Far-C, (**B**) the effect of caspase-3 and (**C**) caspase-9 inhibitor in ameliorating Far-c mediated apoptosis. * *p* < 0.05.

**Figure 7 pharmaceuticals-15-01070-f007:**
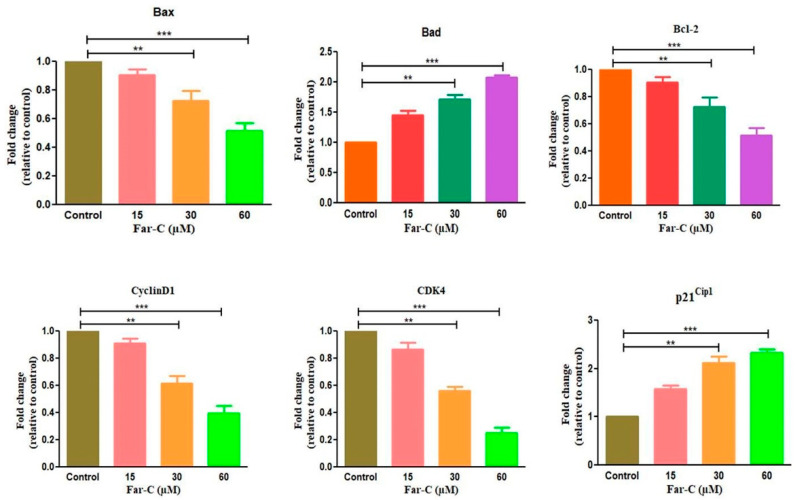
Efficacy of Far-C in modulating the mRNA expression of apoptotic (*Bad*, *Bax*, *Bcl2*) and cell cycle regulatory genes (*CyclinD1*, *CDK4* and *p21^Cip1^*). ** *p* < 0.01, *** *p* < 0.001.

**Table 1 pharmaceuticals-15-01070-t001:** List of primer sequences used during the study.

Gene Name	Forward Sequence	Reverse Sequence
GAPDH	CGACCACTTTGTCAAGCTCA	CCCCTCTTCAAGGGGTCTAC
Bax	GCTGGACATTGGACTTCCTC	CTCAGCCCATCTTCTTCCAG
Bad	CCTCAGGCCTATGCAAAAAG	AAACCCAAAACTTCCGATGG
Bcl2	ATTGGGAAGTTTCAAATCAGC	TGCATTCTTGGACGAGGG
cyclinD1	CTTCCTCTCCAAAATGCCAG	AGAGATGGAAGGGGGAAAGA
CDK4	CCTGGCCAGAATCTACAGCTA	ACATCTCGAGGCCAGTCATC
p21^Cip1^	TGTCCGTCAGAACCCATG	GTGGGAAGGTAGAGCTTGG

## Data Availability

Data is contained within the article.

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
