# Peer review of "Farnesiferol C Exerts Antiproliferative Effects on Hepatocellular Carcinoma HepG2 Cells by Instigating ROS-Dependent Apoptotic Pathway"

_pharmaceuticals, 2022, doi:10.3390/ph15091070_

Round 1

Reviewer 1 Report

The paper highlights new and moderately significant results for the sesquiterpene coumarin natural product Farnesiferol-c (Far-C) in the context of hepatocellular carcinoma, a disease of significant burden and unmet medical need. The study is carried out and written up to a high scientific standard. The major limitation of the study is the exclusive focus on one liver cancer cell line (HepG2); the authors need to explain the features and clinical relevance of this cell line, and the limitations of HepG2 as a model for drug development.

The following points additionally need attention:

Introduction, line 35: should say '>1 million individuals by 2025'.

Introduction, line 58-59: the authors should give more detail on the biological attributes of Far-C (as the subject of the study); and provide the chemical structure of Far-C, alongside the general structure of sesquiterpene coumarins.

Results, line 70: 83.13 NOT 8313.

Page 4, Fig. 3: the figure results need a little more explanation to help the reader (AO and EtBr).

Page 6, Fig. 6: is the Bax mRNA expression plot correct - I think this should be a dose-dependent increase?

Author Response

Response to Reviewer 1 Comments

Comments and Suggestions for Authors

Point 1: The paper highlights new and moderately significant results for the sesquiterpene coumarin natural product Farnesiferol-c (Far-C) in the context of hepatocellular carcinoma, a disease of significant burden and unmet medical need. The study is carried out and written up to a high scientific standard. The major limitation of the study is the exclusive focus on one liver cancer cell line (HepG2); the authors need to explain the features and clinical relevance of this cell line, and the limitations of HepG2 as a model for drug development.

Response 1: The authors would like to state that Hep G2 is an immortal cell line which was derived in 1975 from the liver tissue of a 15-year-old Caucasian male from Argentina with a well-differentiated hepatocellular carcinoma. These cells are epithelial in morphology, have a modal chromosome number of 55, and are not tumorigenic in nude mice. The cells secrete a variety of major plasma proteins such as albumin, and the acute-phase proteins fibrinogen, alpha 2-macroglobulin, alpha 1-antitrypsin, transferrin, and plasminogen. They have grown successfully in large-scale cultivation systems. Hep G2 will respond to stimulation with human growth hormone. Thus, they are a suitable in vitro model system for the study of polarized human hepatocytes. However, they have substantial disadvantages such as rapid in vitro dedifferentiation, limited culture duration of only a couple of days, donor variability, and lack of cell proliferation.

The following points additionally need attention:

Point 2: Introduction, line 35: should say '>1 million individuals by 2025'.

Response 2: Authors’ response: The authors apologize for this typological error and the correction has been done in the revised manuscript.

Point 3: Introduction, line 58-59: the authors should give more detail on the biological attributes of Far-C (as the subject of the study); and provide the chemical structure of Far-C, alongside the general structure of sesquiterpene coumarins.

Response 3: As per the insightful suggestion of the learned reviewer the authors have incorporated the biological attributes of Far-C in the introduction section of the revised manuscript. We have drawn the structure of Far-C and the same has been included in the revised manuscript.

Point 4: Results, line 70: 83.13 NOT 8313.

Response 4: The authors apologize for this typological error and the correction has been done in the revised manuscript.

Point 5: Page 4, Fig. 3: the figure results need a little more explanation to help the reader (AO and EtBr).

Response 5: As per the suggestion of the learned reviewer, the authors have incorporated more explanation in the results of AO and EtBr so that it will get easier for the readers.

Point 6: Page 6, Fig. 6: is the Bax mRNA expression plot correct - I think this should be a dose-dependent increase?

Response 6: The authors apologize for this unintentional error and the correction has been done in the revised manuscript.

Reviewer 2 Report

General properties of Farnesiferol C have been previously published (Hasanzadeh et al 2017, Toxicology reports, Lee et al 2010, Molecular cancer therapeutics). Current manuscript does not add novelty to new findings. Any lead compounds when given at sufficient quantity will have a negative impact on all cell types. This does not necessary translate to meaningful therapeutic. The author should consider using a non-tumorgenic liver cell-line as comparison to show specificity against cancer HCC. Additionally, a better approach to drug characterization would be the use of system approach (RNA seq or proteomic) to determine the overall mechanism of drug function. In doing so, the author will be able to better characterize the actual drug function and perform recovery assay (ie: overexpression of certain gene to confer resistance against Farnesiferol C). Use of anti-apoptotic inhibitor does not constitute as a characterization for drug function. If author has considered the above-mentioned factors, it would raise the quality of the manuscript and add new discovery to their findings.  

Author Response

Response to Reviewer 2 Comments

Comments and Suggestions for Authors

Point 1: General properties of Farnesiferol C have been previously published (Hasanzadeh et al 2017, Toxicology reports, Lee et al 2010, Molecular cancer therapeutics). Current manuscript does not add novelty to new findings. Any lead compounds when given at sufficient quantity will have a negative impact on all cell types. This does not necessary translate to meaningful therapeutic. The author should consider using a non-tumorgenic liver cell-line as comparison to show specificity against cancer HCC. Additionally, a better approach to drug characterization would be the use of system approach (RNA seq or proteomic) to determine the overall mechanism of drug function. In doing so, the author will be able to better characterize the actual drug function and perform recovery assay (ie: overexpression of certain gene to confer resistance against Farnesiferol C. Use of anti-apoptotic inhibitor does not constitute as a characterization for drug function. If author has considered the above-mentioned factors, it would raise the quality of the manuscript and add new discovery to their findings.

Response 1: In response to the concern raised by the learned reviewer, the authors would like to state that there are only two reports of Far-C demonstrating anticancer efficacy on lung cancer [Jung, J. H., Kim, M. J., Lee, H., Lee, J., Kim, J., Lee, H. J., Shin, E. A., Kim, Y. H., Kim, B., Shim, B. S., & Kim, S. H. (2016). Farnesiferol c induces apoptosis via regulation of L11 and c-Myc with combinational potential with anticancer drugs in non-small-cell lung cancers. Scientific reports, 6, 26844. https://doi.org/10.1038/srep26844] and breast cancer [Hasanzadeh, D., Mahdavi, M., Dehghan, G., & Charoudeh, H. N. (2017). Farnesiferol C induces cell cycle arrest and apoptosis mediated by oxidative stress in MCF-7 cell line. Toxicology reports, 4, 420–426. https://doi.org/10.1016/j.toxrep.2017.07.010]. Till date, there is no such report demonstrating the chemopreventive potential of Far-C on hepatocellular carcinoma. The authors did not completely agree with the statement that “any lead compounds when given at sufficient quantity will have a negative impact on all cell types” because it is not mandatory that an anticancer drug will certainly elicit similar anticancer responses in different carcinoma cell lines. This can also be explained on the basis of features and clinical attributes of various types of cell lines. In this case, we are differentiating HepG2 cells from A549 and MCF-7 cells. Human HepG2 cells have a low metastatic phenotype. The TP53 gene can be found in liver cancer in two types: mutant and wild. Wild-type TP53 is observed in the HepG2 cell line, as in HCC and HB. In HCC, mutant TP53 tumors have higher malignant potentials than those with wild-type TP53. The TP53 gene is critical in suppressing cancer in humans, as it plays a role in cell cycle arrest, apoptosis, and ageing. Thus, this mutation in the gene can promote cell proliferation. Whereas MCF-7 is a poorly-aggressive and non-invasive cell line, normally being considered to have low metastatic potential. However, A549 is an epithelial carcinoma derived from a 58 year old male patient, known to be KRAS mutant and EGFR wild type. A549 cells are relatively large cells, with a doubling time of approximately 24 hours. They are suitable for in vitro and in vivo experimentation. Immunocompromised mice should be used for in life studies, and will form tumors and metastases following implantation of the cells. Therefore, on the basis of above-stated comparisons between the three cell lines, we can say that it is not mandatory that every anticancer drug will exhibit similar anticancer efficacy against different cancer cell lines.

Furthermore, the authors agree with the learned reviewer that we should use a non-tumorigenic liver cell-line as comparison to show specificity against cancer HCC and using RNA seq or proteomics for better drug development. However, the authors would like to state that this is mere a preliminary report demonstrating antiproliferative activity of Far-C on HepG2 cells and due to some financial constraints it will not be possible for us to perform these experiments. The authors request the learned reviewer to understand our position and consider the remarks made herein.

Reviewer 3 Report

The research article entitled "Farnesiferol C exerts antiproliferative effects on hepatocellular 2 carcinoma HepG2 cells via instigating ROS dependent apop- 3 totic pathway"the authors tested 13 the hypothesis that Far-c instigated oxidative stress resulted in anti-proliferation and apoptosis in- 14 stigation within human liver cancer HepG2 cells. The observations reported herewith indicated that 15 Far-C exerted considerable cytotoxic effects on HepG2 cells by reducing the cell viability (p<0.001) 16 in a dose-dependent manner. Far-C exposure also resulted in enhanced ROS production (p<0.01) 17 which subsequently led to loss of mitochondrial membrane potential. Far-C instigated oxidative 18 stress also led to enhanced nuclear fragmentation and condensation as revealed through Hoechst- 19 33342. These molecular changes post-Far-C exposure also incited apoptotic cell death which con- 20 comitantly changed lead to significant activation of caspase-3 (p<0.001). Furthermore, Far-C also 21 exhibited its competence in altering the expression of genes involved in apoptosis regulation (Bax, 22 Bad and Bcl2) along with genes exerting regulatory effects on cell cycle (cyclinD1) and its progres- 23 sion (p21Cip1 and CDK4). The evidences thus provided clearly the preclinical efficacy of Far-C 24 against HepG2 cells however further mechanistic investigations deciphering the alteration of differ- 25 ent pathways post-Far-C exposure will be highly beneficial.

It requires minor revision which is like a major one,

1-what is meant by Farnesiferol C exerts, the compound Farnesiferol C isolated as pure compound?

2-Plant material should be included.

Author Response

Response to Reviewer 3 Comments

Comments and Suggestions for Authors

The research article entitled "Farnesiferol C exerts antiproliferative effects on hepatocellular carcinoma HepG2 cells via instigating ROS dependent apoptotic pathway" the authors tested the hypothesis that Far-c instigated oxidative stress resulted in anti-proliferation and apoptosis instigation within human In liver cancer HepG2 cells. The observations reported herewith indicated that 15 Far-C exerted considerable cytotoxic effects on HepG2 cells by reducing the cell viability (p<0.001) 16 in a dose-dependent manner. Far-C exposure also resulted in enhanced ROS production (p<0.01) 17 which subsequently led to the loss of mitochondrial membrane potential. Far-C instigated oxidative 18 stress also led to enhanced nuclear fragmentation and condensation as revealed through Hoechst- 33342. These molecular changes post-Far-C exposure also incited apoptotic cell death which con- 20 comitantly changed lead to significant activation of caspase-3 (p<0.001). Furthermore, Far-C also 21 exhibited its competence in altering the expression of genes involved in apoptosis regulation (Bax, 22 Bad and Bcl2) along with genes exerting regulatory effects on cell cycle (cyclinD1) and its progression (p21Cip1 and CDK4). The evidence thus provided clearly the preclinical efficacy of Far-C 24 against HepG2 cells however further mechanistic investigations deciphering the alteration of different pathways post-Far-C exposure will be highly beneficial.

It requires minor revision which is like a major one,

Point 1: What is meant by Farnesiferol C exerts, the compound Farnesiferol C isolated as a pure compound?

Response 1: The authors would like to state that we have purchased Farnesiferol-C as a purified compound from Sigma Aldrich, USA and the same information has been incorporated in the revised manuscript.

Point 2: Plant material should be included.

Response 2: The authors would like to state that we have used a purified compound (Farnesiferol-C) from Sigma Aldrich, USA and the same information has been incorporated in the revised manuscript.

Round 2

Reviewer 3 Report

accept in present form